suicide; global mental health; LGBTQ+ youth; Peru; minority stress

**Corresponding author:**
Juan C. Jauregui;
Email: jjauregui@g.ucla.edu

# Understanding suicide risk among LGBTQ+ youth in Peru: Findings from a nationwide mental health survey

Juan C. Jauregui[1] [ID], Michael Reyes-Diaz[2], Fran León-Morris[2], Ronita Nath[3] [ID], Ashley B. Taylor[3] [ID] and Kelika A. Konda[4] [ID]

[1]Department of Social Welfare, University of California Los Angeles, Los Angeles, USA; [2]Investigaciones Médicas en Salud, Lima, Peru; [3]The Trevor Project, West Hollywood, USA and [4]Department of Population and Public Health Sciences, USC Keck School of Medicine, Los Angeles, USA

## Abstract

LGBTQ+ youth globally face increased suicide risk, yet evidence from Latin America, particularly Peru, is limited. Understanding factors influencing suicidality among LGBTQ+ youth in Peru is essential for developing culturally relevant interventions. This study analyzed data from The Trevor Project's 2024 Peru National Survey on the Mental Health of LGBTQ+ Young People (N = 4,643; age 14–24, mean age = 17.34). Logistic regression models examined associations between suicide-related outcomes (lifetime and past-year suicidal ideation, past-year suicide attempts), and positive screens for depressive and anxiety symptoms, mental healthcare desire, sexual orientation "outness" and perceived community acceptance. Approximately 73.5% reported lifetime suicidal ideation, 55.0% past-year ideation and 37.1% past-year attempts. Positive screens for depressive and anxiety symptoms were associated with higher odds of suicidality (aOR range: 1.80–2.88). Compared to youth who did not desire care, both unmet (aOR range: 1.64–2.16) and met (aOR range: 1.26–1.36) mental healthcare desire were associated with higher odds of suicidality, with unmet need showing stronger effects. Higher outness (aOR range: 1.56–1.75), younger age (14–17 years), and gender diversity were associated with higher odds of suicidality. Findings highlight the urgent need for accessible, identity-affirming mental health interventions in Peru, tailored to developmental stages and responsive to the stressors faced by LGBTQ+ youth.

## Impact Statements

This study provides national-level evidence on suicide risk among LGBTQ+ youth in Peru, a population with urgent and underrecognized mental health needs in a context of limited resources, pervasive stigma and scarce data. Our findings reveal alarmingly high rates of suicidal ideation and attempts, with over one in three LGBTQ+ youth reporting a suicide attempt in the past year. Critically, we show that risk is associated not only with mental health symptoms such as depression and anxiety, but also with structural and developmental factors, including unmet needs for mental health care, lack of community support and the complex realities of disclosing one's LGBTQ+ identity in unaccepting environments. By highlighting how specific subgroups (i.e., minors, transgender and non-binary youth, and those more publicly "out") face heightened vulnerability, this research offers concrete targets for intervention. The evidence points to an urgent need for culturally affirming mental health services in Peru that are responsive to both age and identity. These services should equip younger adolescents with coping strategies, ensure gender-affirming care and build safe, stigma-free environments in families, schools and communities. Although centered on Peru, the implications extend across Latin America, where LGBTQ+ youth often face similar structural barriers, social exclusion and gaps in mental health infrastructure. This study advances the field of global mental health by generating actionable evidence to guide policy, inform community-based programming and strengthen suicide prevention strategies tailored to LGBTQ+ youth in resource-constrained and stigmatizing settings.

## Introduction

Peru's lesbian, gay, bisexual, transgender and queer (LGBTQ+) populations navigate elevated risks of poor mental health and suicide within a context of systemic barriers. The national mental health system is marked by fragmentation and large regional disparities in service availability and quality of care (Carrasco-Escobar et al., 2020; Arriola-Vigo and Diez-Canseco, 2021). These inequities are intensified by a severe shortage of trained mental health professionals, with only 9.51 psychologists and 2.95 psychiatrists per 100,000 residents (World Health Organization, 2020). Access to care is further constrained by social conditions that discourage help-seeking and





undermine service delivery. Pervasive stigma toward LGBTQ+ people, reflected in national surveys showing that 59% of Peruvians oppose same-sex marriage and nearly one-third report discomfort with having a gay neighbor (Instituto de Estudios Peruanos, 2019), can foster discriminatory treatment in health care and discourage disclosure of sexual orientation or gender identity to providers (Hatzenbuehler, 2016; Brooks et al., 2018). The absence of anti-discrimination protections, alongside policy actions such as the 2024 classification of transgender identity as a mental disorder (Kottke, 2024), reinforces exclusion and legitimizes prejudice. In this context, nearly two-thirds of LGBTQ+ Peruvians report lifetime exposure to violence or discrimination (INEI, 2018), while harassment, threats and harmful practices such as conversion "therapy" remain common (León-Morris et al., 2024).

Globally, suicide is among the leading causes of death in young people (World Health Organization, 2025). Adolescence and early adulthood are developmental stages marked by limited autonomy, high dependence on parents or caregivers and heightened sensitivity to social evaluation, all of which contribute to increased vulnerability (Arnett, 2000; Sisk and Gee, 2022). For LGBTQ+ youth, these developmental risks are compounded by disproportionate exposure to school-based victimization, adverse childhood experiences and identity-based discrimination (Cox et al., 2011; Gorse, 2020). Research from high-income countries consistently shows that LGBTQ+ young people are more than twice as likely to attempt suicide as their heterosexual and cisgender peers (King et al., 2008). Yet, evidence from low- and middle-income countries (LMICs), including those in Latin America, remains limited, even though service gaps, structural stigma and the absence of protective legislation may exacerbate risk (Prince et al., 2007; Patel et al., 2018; Jauregui et al., 2025).

Minority stress theory provides a useful framework for understanding the elevated suicide risk and mental health disparities faced by LGBTQ+ youth. The theory posits that identity-based stigma, discrimination and structural exclusion generate chronic stress that harms mental health, while affirming environments and coping resources can buffer these effects (Meyer, 2003; Frost et al., 2015; Mongelli et al., 2019). In Peru, however, these identity-based stressors are compounded by poverty and educational inequities, which can limit the ability to afford care, travel to urban service hubs or access providers trained to meet LGBTQ+ needs (Tzenios, 2019; Carrasco-Escobar et al., 2020; Orozco-Poore et al., 2024). Evidence from transgender adolescents and young women in Lima illustrates how even protective factors such as strong community networks, legal advocacy and health-promoting behaviors are constrained by pervasive transphobia, family rejection and sexual violence, perpetuating cycles of harm (Orozco-Poore et al., 2024). Similar patterns appear in other health domains, where structural stigma and resource gaps undermine well-being (Hatzenbuehler and Pachankis, 2016; Reisner et al., 2017; Newman et al., 2023).

Understanding how these structural and psychosocial factors interact to shape suicide risk is central to advancing global mental health research and practice, particularly for marginalized youth. Guided by minority stress theory, this study analyzes data from The Trevor Project's Peru National Survey on the Mental Health of LGBTQ+ Young People to examine how mental health symptoms, access to care, outness and perceived community acceptance relate to suicidal ideation and attempts. By situating these factors within Peru's broader social, legal and service-delivery context, the study generates evidence to inform suicide prevention strategies that are culturally grounded, identity-affirming and developmentally appropriate.

## Methods

### Study design and data source

We performed a cross-sectional secondary analysis using data from The Trevor Project's 2024 Peru National Survey on the Mental Health of LGBTQ+ Youth, administered online from October 24 to December 12, 2022. The survey was adapted from The Trevor Project's US-based instrument and translated into Spanish with cultural tailoring for the Peruvian context. Adaptation involved iterative review by local researchers and a community advisory board of LGBTQ+ youth in Peru, who vetted recruitment text and images as well as survey wording to ensure cultural and linguistic appropriateness prior to launch.

Recruitment occurred through targeted advertisements on Facebook and Instagram, which directed users to a secure study webpage. At the time of recruitment, platform policies did not permit targeting advertisements based on sexual orientation, gender identity or age below the age of majority. When referring to "targeted" advertisements, we mean that the advertisement text and imagery were designed to signal the intended study population (e.g., LGBTQ+ youth aged 14–24 years), allowing individuals to self-identify as potentially eligible and voluntarily click through to learn more about the study. There, participants reviewed study information, completed eligibility screening and provided informed consent (informed assent for minors). Parental consent was waived given that requiring permission could pose a greater risk, such as unintentional disclosure of participants' LGBTQ+ identity, than allowing confidential participation. To protect privacy, the survey was confidential and did not collect identifying information such as names or contact details.

The survey assessed demographics, mental health (including suicidality), access to mental healthcare and sexual health services, stigma and violence (e.g., threats, physical harm, discrimination, conversion "therapy" and housing instability), and protective factors (e.g., community acceptance, supportive relationships, affirming spaces and outness). Information and links to free emotional support services were provided throughout the survey, particularly adjacent to suicidality-related items. Respondents did not receive an incentive to participate. The study was approved by the Comité Institucional de Ética en Investigación (CIEI) de Investigaciones Médicas en Salud (INMENSA) (No. 0012–2022 CIEI) in Peru and Solutions Institutional Review Board in the United States (Protocol Number 285).

### Study sample

Eligible participants were LGBTQ+ youth aged 14–24 living in Peru who self-identified with a sexual orientation other than heterosexual, a gender identity other than cisgender or both. Participants were required to complete a mid-survey validation item to ensure attentiveness, and responses that were incomplete or invalid were excluded from the analytic dataset. During data collection, a temporary quota was implemented to limit further enrollment of participants assigned female at birth (AFAB) in order to maintain demographic balance across sex assigned at birth. Beginning December 2, 2022, respondents who indicated female sex assigned at birth and entered the survey after the quota was reached were automatically redirected to the end of the survey with a message indicating that they were not eligible to continue at that time and were not included in the analytic sample. In addition, a small number of respondents who indicated residence outside of Peru

were screened out at eligibility and similarly informed that they were not eligible and thanked for their time.

### Measures

The primary outcomes were three binary indicators of suicidality, adapted from the US Centers for Disease Control and Prevention's Youth Risk Behavior Surveillance System (YRBS) (Underwood et al., 2020). Lifetime suicidal ideation was measured by the question "Have you ever seriously considered suicide?" (yes/no). Participants who responded "no" skipped subsequent questions on past-year ideation and suicide attempts but were coded as "no" on those items for analytic purposes, as this was a logical assumption. Past-year suicidal ideation was measured with the question "In the last 12 months, have you seriously considered suicide?" (yes/no). Similarly, those responding "no" skipped the suicide attempt item but were coded as "no" for attempts. Past-year suicide attempt was assessed with the item "In the last 12 months, how many times did you attempt suicide?" Response options included "1 time," "2 or 3 times," "4 or 5 times," "6 or more times" and "Not in the last year." Participants reporting one or more attempts were coded as "yes," and those selecting "Not in the last year" (or who had skipped due to earlier responses) were coded as "no."

Key independent variables included symptoms of depression, symptoms of anxiety, past-year mental healthcare desire, sexual orientation disclosure (outness) and perceived community acceptance. Depression symptoms were measured using the validated two-item Patient Health Questionnaire (PHQ-2), and anxiety symptoms were measured with the two-item Generalized Anxiety Disorder (GAD-2) screening tool (Löwe et al., 2005; Plummer et al., 2016); for both measures, scores range from 0 to 6, with a score of 3 or higher indicating a positive screen for elevated depressive or anxiety symptoms warranting further assessment.

Past-year mental healthcare desire was measured with the question "In the last 12 months, have you sought psychological or emotional support from a counselor or mental health professional?", which was categorized as no desire ("no"), unmet desire ("yes, but I didn't get it") and desire met ("yes, and I got it"). For participants wanting but not receiving mental health care, barriers to access were assessed with the question "Did you not see a counselor or mental health professional for any of the following reasons?" Response options included financial constraints, logistical barriers, stigma-related concerns and prior negative experiences, among others. Participants could select multiple responses.

Outness regarding sexual orientation was measured by asking participants to indicate the extent to which they were "out" to people they knew; responses were grouped into three categories: not out ("none of the people I know"), moderately out ("a few people I know" or "some of the people I know") and very out ("a lot of the people I know" or "all of the people I know"). Perceived community acceptance was measured by asking "How accepted are LGBTQ+ people in the community where you currently live?"; the four-point scale was collapsed into two categories: accepting (somewhat or very accepting) and unaccepting (somewhat or very unaccepting).

Demographic variables of interest were age group and gender identity. Age was grouped into minors (14–17 years) and young adults (18–24 years). Gender identity was categorized as cisgender, transgender/non-binary or questioning.

Covariates included geographic region, migrant status, school enrollment, employment status and socioeconomic status. Geographic regions were categorized as five macro-regions: Lima/Callao (Metropolitan Lima or Callao), north (Ancash, Cajamarca, La Libertad, Lambayeque, Piura, Tumbes), south (Apurímac, Arequipa, Cusco, Moquegua, Puno, Tacna), central (Ayacucho, Huancavelica, Huánuco, Ica, Junín, Lima provincial, Pasco) and jungle (Amazonas, Loreto, Madre de Dios, San Martín, Ucayali).

### Statistical analysis

We calculated descriptive statistics for sample demographics, suicide-related outcomes and key independent variables. Separate logistic regression models were estimated for each outcome (lifetime ideation, past-year ideation, past-year attempt). Models included all key independent variables (symptoms of depression, symptoms of anxiety, past-year mental healthcare desire, outness and community acceptance), along with demographic variables of interest (age group, gender identity). Models were further adjusted for region, migrant status, school enrollment, employment and socioeconomic status to account for potential confounding. Analyses were conducted on complete case data, with participants missing responses on model variables excluded using listwise deletion; missingness was generally low, and no imputation procedures were applied. Because three distinct models were estimated, the potential for inflated type I error rates should be acknowledged. This analysis is therefore considered exploratory, and findings should be interpreted with caution. Results are reported as adjusted odds ratios (aOR) with 95% confidence intervals. Analyses were conducted in R Studio (version 2021.09.0).

### Results

#### Sample characteristics

A total of 12,425 people began the survey, of whom 7,487 met initial eligibility criteria. After excluding incomplete or invalid responses, the final analytic sample comprised 4,643 LGBTQ+ youth in Peru aged 14–24. Figure 1 summarizes eligibility screening, data cleaning and the validation process.

Nearly half identified as bisexual (48.9%), followed by gay or lesbian (28.0%) and pansexual (12.1%); smaller proportions identified as questioning or unsure (4.6%), asexual (2.8%), or queer (2.5%). Lifetime suicidal ideation was reported by 73.5%, past-year suicidal ideation by 55.0% and past-year suicide attempts by 37.1%. Of those having a desire for mental health care in the past year (N = 2,089), nearly half (45.4%) were unable to access it. Participant demographics are presented in Table 1.

#### Logistic regression analyses

Multivariable logistic regression models identified several psychosocial and structural factors significantly associated with suicide-related outcomes (Table 2). Positive screens for depressive and anxiety symptoms emerged as strong correlates of suicidality, with both symptom screens associated with nearly double the odds of lifetime and past-year suicidal ideation and attempts. Unmet mental healthcare needs were independently associated with higher odds of suicidality across all outcomes. Youth whose needs were met also had higher odds of suicidality compared to those who did not desire care, although effect sizes were smaller. Minors (aged 14–17) showed significantly higher odds of ideation and attempts compared to young adults (ages 18–24), and transgender/non-binary and questioning youth had higher odds of all suicide-related outcomes relative to cisgender peers. Greater outness about sexual

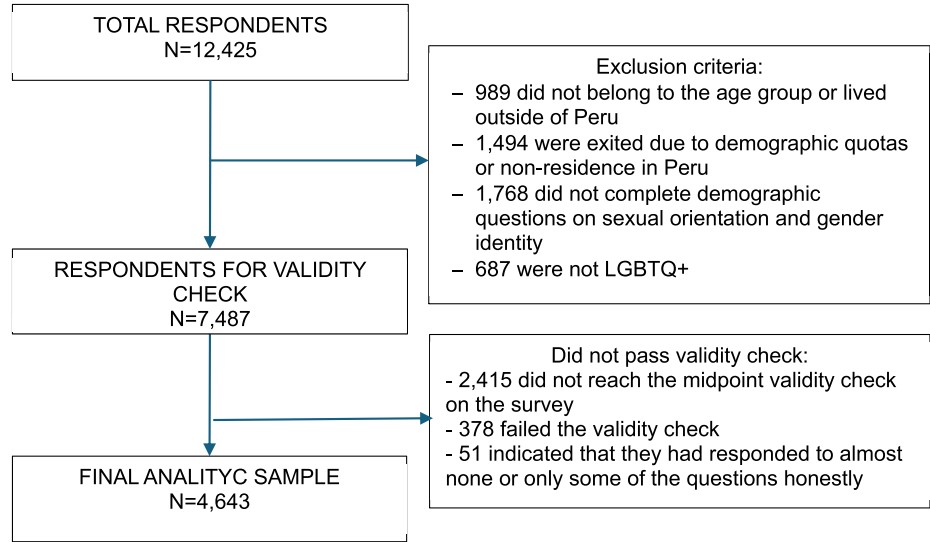

**Figure 1.** Validation and data cleaning process.

**Table 1.** Participant demographics by age group

|  | Total | Minors 14–17 years | Young adults 18–24 years |
|---|---|---|---|
|  | *N* (%) | *N* (%) | *N* (%) |
|  | 4,643 | 2,901 (62.5) | 1,742 (37.5) |
| Sex assigned at birth |  |  |  |
| Male | 1,728 (37.6) | 786 (27.4) | 942 (54.5) |
| Female | 2,081 (62.4) | 2,081 (72.6) | 787 (45.5) |
| Gender identity |  |  |  |
| Cisgender | 3,105 (69.3) | 1,804 (64.8) | 1,301 (76.5) |
| Questioning if transgender or non-binary | 654 (14.6) | 480 (17.3) | 174 (10.2) |
| Transgender or non-binary | 723 (16.1) | 498 (17.9) | 225 (13.2) |
| Region of residence |  |  |  |
| Metropolitan Lima and Callao | 2,054 (44.2) | 1,251 (43.1) | 803 (46.1) |
| North | 859 (18.5) | 535 (18.4) | 324 (18.6) |
| South | 683 (14.7) | 414 (14.3) | 269 (15.4) |
| Central | 787 (17.0) | 530 (18.3) | 257 (14.8) |
| Jungle | 260 (5.6) | 171 (5.9) | 89 (5.1) |
| Enrolled in school/academy/university/ institute |  |  |  |
| Yes | 4,030 (88.0) | 2,708 (94.4) | 1,322 (77.4) |
| No | 548 (12.0) | 161 (5.6) | 387 (22.6) |
| Currently working |  |  |  |
| Yes | 1,085 (23.9) | 346 (12.2) | 739 (43.1) |
| No | 3,463 (76.1) | 2,488 (87.8) | 975 (56.9) |
| Lifetime suicidal ideation |  |  |  |
| Yes | 3,218 (73.5) | 2,075 (75.8) | 1,143 (69.5) |
| No | 1,164 (26.5) | 661 (24.2) | 502 (30.5) |
| Past-year suicidal ideation |  |  |  |
| Yes | 2,344 (55.0) | 1,621 (61.0) | 723 (45.0) |
| No | 1,920 (45.0) | 1,036 (39.0) | 884 (55.0) |

(*Continued*)

**Table 1.** (*Continued*)

| | Total | Minors 14–17 years | Young adults 18–24 years |
|---|---|---|---|
| | *N* (%) | *N* (%) | *N* (%) |
| Past-year suicide attempt | | | |
| Yes | 1,539 (37.1) | 1,131 (44.1) | 408 (25.8) |
| No | 2,605 (62.9) | 1,433 (55.9) | 1,172 (74.2) |

**Table 2.** Logistic regression results for predictors of lifetime and past-year suicidality among LGBTQ+ youth

| | Lifetime ideation aOR (95% CI) | Past-year ideation aOR (95% CI) | Past-year attempt aOR (95% CI) |
|---|---|---|---|
| Positive depression screen (PHQ–2 ≥ 3) | | | |
| No (ref) | – | – | – |
| Yes | **2.88 (2.40–3.46)** | **2.58 (2.19–3.05)** | **2.28 (1.91–2.73)** |
| Positive anxiety screen (GAD–2 ≥ 3) | | | |
| No (ref) | – | – | – |
| Yes | **1.80 (1.50–2.16)** | **2.11 (1.79–2.49)** | **1.91 (1.61–2.27)** |
| Sexual orientation disclosure | | | |
| Not out (ref) | – | – | – |
| Somewhat out | **1.75 (1.35–2.26)** | **1.39 (1.09–1.78)** | 1.28 (0.99–1.66) |
| Out to most or all | **1.56 (1.13–2.17)** | 1.27 (0.93–1.74) | 1.23 (0.89–1.71) |
| Mental healthcare desire | | | |
| Did not desire (ref) | – | – | – |
| Unmet mental health desire | **2.16 (1.69–2.78)** | **1.97 (1.60–2.42)** | **1.64 (1.34–2.01)** |
| Met mental health desire | **1.36 (1.11–1.68)** | **1.35 (1.12–1.63)** | **1.26 (1.04–1.53)** |
| Age group | | | |
| Young adults 18–24 (ref) | – | – | – |
| Minors 14–17 | **1.26 (1.05–1.52)** | **1.89 (1.39–2.24)** | **2.27 (1.89–2.73)** |
| Gender identity | | | |
| Cisgender (ref) | – | – | – |
| TGNB identity | **2.48 (1.87–3.35)** | **1.88 (1.50–2.36)** | **1.74 (1.41–2.16)** |
| Questioning TGNB | **1.85 (1.41–2.46)** | **1.76 (1.39–2.24)** | **1.64 (1.31–2.06)** |
| Perceived community acceptance | | | |
| Unaccepting (ref) | – | – | – |
| Accepting | 1.02 (0.85–1.21) | 0.94 (0.80–1.11) | 1.00 (0.85–1.18) |

*Note:* Boldface values indicate statistical significance, *p*-value<0.05.

orientation was associated with higher odds of lifetime suicidal ideation compared to those who were not out. Perceived community acceptance was not significantly associated with any suicide-related outcomes.

### Barriers to accessing mental health support

Among participants who reported wanting mental health care in the past year but did not receive it (N = 949), the most frequently cited barrier was the belief that providers would not understand their sexual orientation or gender identity (73.1%). Nearly half (45.7%) said they could not afford care, and over one-third reported fear of talking about mental health concerns with someone else (38.0%) or worry about not being taken seriously (34.3%). Stigma-related fears were also common: 31.0% were not out about their LGBTQ+ identity and feared being outed, 28.6% were distrustful that treatment would work, and 28.2% feared providers would attempt to repress their LGBTQ+ identity. A quarter (25.3%) did not want to seek parental or caregiver permission, and 19.1% reported being explicitly denied permission. Other barriers

included reluctance to receive virtual care at home (18.3%), concerns that providers would focus only on their sexual orientation or gender identity (17.2%), prior negative experiences (14.7%), transportation challenges (12.5%) and lack of available providers (4.3%).

## Discussion

This study reveals alarmingly high rates of suicidality among LGBTQ+ youth in Peru, with nearly three-quarters reporting lifetime suicidal ideation, over half reporting ideation in the past year and more than one in three reporting a suicide attempt in the past year. These figures are substantially higher than global averages for general youth populations, where pooled estimates place lifetime suicidal ideation at 18%, past-year ideation at 14.2% and past-year suicide attempts at 4.5% (Lim et al., 2019). National data from Peru's Global School-based Health Survey (GSHS) similarly show that nearly 20% of adolescents reported past-year suicidal ideation and 12.7% had formed a suicide plan in the past year (Hernández-Vásquez et al., 2019), underscoring the disproportionate burden borne by LGBTQ+ youth in Peru relative to their peers. While elevated suicide risk among LGBTQ+ youth is well documented internationally (Miranda-Mendizábal et al., 2017; Hatchel et al., 2021), the magnitude observed in this national sample of Peruvian youth suggests an urgent need for intervention. These findings should be interpreted in the broader context of Peru's underdeveloped mental health infrastructure, pervasive stigma toward LGBTQ+ populations and lack of legal protections, which together may contribute to heightened vulnerability and the persistently high burden of suicidality observed in this group.

Consistent with a large body of global evidence, screening-level depressive and anxiety symptoms, as measured by brief self-report instruments, were strongly associated with lifetime and past-year suicidal ideation and attempts in this sample. It is important to note that these measures indicate elevated symptom burden rather than clinical diagnoses, as participants did not undergo diagnostic assessment. These symptom domains may influence suicidal behaviors in distinct ways; while depression is among the strongest predictors of suicidal ideation, anxiety has been more closely linked to the transition from ideation to attempts, especially when co-occurring with mood disorders (Sareen et al., 2005; Gili et al., 2019; Fang et al., 2024). Given that roughly half of severe mental disorders emerge by age 14 and 75% before age 24 (Kessler et al., 2005), early identification and treatment are critical for suicide prevention.

These patterns are particularly concerning in the Peruvian context, where an estimated 69–85% of individuals with mental health needs do not receive care and where 85% of psychiatrists are concentrated in Lima, leaving rural and underserved regions with minimal coverage (Toyama et al., 2017). Although all participants in this study were provided with information about available services, these resources remain limited in reach and capacity, and anecdotal reports suggest demand increased during the survey period. Notably, our models found no significant differences by geographic region, suggesting that elevated suicide risk extends beyond Lima to the wider country, a critical finding given that most prior research on LGBTQ+ mental health has been confined to the capital. Expanding youth-focused detection and intervention, particularly through culturally responsive, LGBTQ-inclusive approaches, may therefore be a key component of suicide prevention. However, screening efforts are unlikely to be effective unless paired with affirming supports and a trained mental health workforce capable of meeting LGBTQ+ youths' needs.

Unmet and met desire for mental health care were independently associated with significantly higher odds of suicidality compared to youth who did not report a desire for care, although the magnitude of these associations was consistently larger among those with unmet mental health needs. This graded pattern suggests that youth who perceive a need for care but are unable to access or receive appropriate services may experience particularly elevated levels of distress, while also underscoring that service contact alone may be insufficient to fully mitigate suicide risk. The most frequently reported barrier was the perception that providers would not understand participants' sexual orientation or gender identity, alongside financial constraints, fear of discussing mental health concerns, worry about not being taken seriously, apprehension about being outed, and distrust in treatment effectiveness or concern that providers might attempt to suppress their LGBTQ+ identity. These findings align with prior research showing that stigma, limited mental health literacy and preferences for self-reliance often deter young people from seeking help (Gulliver et al., 2010; Clement et al., 2015), while also highlighting LGBTQ+-specific barriers, such as fear of being outed, that have been documented as especially salient deterrents for LGBTQ+ youth (McDermott et al., 2018; Chong et al., 2021). Such barriers may delay treatment initiation and erode trust in the healthcare system over time, contributing to persistent mental health disparities. At the same time, these findings must be interpreted cautiously given the cross-sectional design. Reverse causation is plausible, particularly in relation to service use and unmet need, as youth experiencing more severe suicidal ideation or attempts may be more likely to seek care, perceive a need for services or report dissatisfaction with available support. In this context, unmet mental health need may reflect the intensity or complexity of underlying distress rather than a causal driver of suicidality. Addressing these gaps therefore requires more than simply expanding the supply of services; it necessitates embedding identity-affirming practices across points of care, training providers to deliver culturally competent and stigma-free services and engaging families and caregivers in acceptance-based interventions. Evidence suggests that LGBTQ-affirming services and family-focused supports are associated with better mental health outcomes and lower suicide risk (Ryan et al., 2010; Turban et al., 2022). Scalable approaches such as school-based screening, integration in primary care and task-sharing with trained non-specialist providers offer promising avenues for closing treatment gaps if implemented with explicit protections for LGBTQ+ youth (Hoeft et al., 2018; Roxanne et al., 2018).

Minors in this study had markedly higher odds of both past-year suicidal ideation and attempts compared to young adults. This developmental vulnerability may reflect lower autonomy and decision-making capacity, greater dependence on potentially non-affirming family members or school environments, and fewer accessible coping resources (Cox et al., 2011). Adolescents may also be in earlier stages of identity development, making them more susceptible to the psychological effects of rejection, victimization and concealment stress (Brown and Trevethan, 2010). These findings underscore the urgent need for age-specific suicide prevention strategies that address the unique developmental, social and identity-related challenges facing LGBTQ+ adolescents. While less is known about suicidality among younger LGBTQ+ populations, continued attention to minors and preteens is important for building prevention strategies that reflect developmental needs across the life course. As with other findings

in this study, these associations should not be interpreted as evidence of temporal or causal effects; rather, they highlight subgroups with a disproportionate burden of suicidality that warrant focused prevention and support.

Transgender, non-binary and questioning youth had between 1.68- to 2.22-times odds of all suicide-related outcomes compared to cisgender peers. This elevated risk aligns with minority stress theory, which posits that stigma, discrimination and identity-based stressors create chronic psychological burdens that increase suicide risk. Notably, a larger proportion of minors in our sample identified as transgender, non-binary or questioning compared to young adults, reflecting broader global trends in which younger cohorts are increasingly embracing diverse gender identities (Wittlin et al., 2023; Herman and Flores, 2025).

In Peru, structural inequities compound these risks. There is no legal framework for gender identity recognition, preventing individuals from aligning official documents with their lived gender, a barrier that has been shown to exacerbate stress and limit access to essential services (Perez-Brumer et al., 2017; Reisner et al., 2021). Gender-diverse youth also face disproportionate exposure to violence, social exclusion and discrimination in homes, schools and communities (INEI, 2018; Romani et al., 2021; Silva-Santisteban et al., 2024). The situation is particularly dire for transgender women, whose average life expectancy across Latin America is estimated at only 35 years, with homicide being a leading cause of death (Inter-American Commission on Human Rights, 2015). Such conditions reflect a process of dehumanization that renders gender-minority populations socially and politically vulnerable and, in extreme cases, physically unsafe (Rodríguez Madera, 2022). Recent policy actions in Peru, including the classification of transgender identity as a mental disorder (Kottke, 2024), have reinforced institutional stigma. More broadly, the current political climate, where leaders in multiple regions openly deny the existence of gender diversity, risks emboldening hostility and granting social legitimacy to prejudice, thereby worsening mental health outcomes for LGBTQ+ youth. These realities highlight the urgent need for protective legislation, the depathologization of gender diversity, the expansion of gender-affirming care and explicit inclusion of gender diversity in clinical services, educational policies and community programs.

Greater disclosure of sexual orientation was associated with higher odds of lifetime ideation. While we did not directly assess the contexts in which disclosure occurred, prior research suggests that the effects of disclosure vary depending on the surrounding environment. Minority stress and disclosure process models propose that "coming out" can improve well-being when met with acceptance, but in stigmatizing environments, it can increase exposure to rejection, harassment and violence (Chaudoir Chaudoir and Fisher, 2010; Meyer, 2003; Toomey et al., 2010). Studies also show that authenticity and supportive climates buffer risk, whereas hostile family or school climates amplify it (Ryan et al., 2009; Legate et al., 2012). Suicide prevention should therefore support youths' autonomy around disclosure, prioritize safety planning and build affirming environments in families, school and clinics through anti-bullying policies, staff training and family-acceptance interventions (Kosciw et al., 2022). Importantly, "coming out" is not universally necessary; individualized, developmentally sensitive approaches that center safety and choice are warranted. It is also possible that youth experiencing suicidal distress are more likely to disclose their sexual orientation in search of support, underscoring again the potential for bidirectional relationships that cannot be disentangled in cross-sectional data.

Perceived community acceptance was not significantly associated with suicidality in this study, a finding that should be interpreted cautiously. Community acceptance was assessed using a single, global self-report item, which was dichotomized and may not have captured the variability across specific interpersonal and institutional contexts most salient to LGBTQ+ youths' mental health, such as family acceptance, school climate or peer support. Prior research suggests that broader community attitudes may influence suicide risk indirectly or conditionally, operating through these more proximal environments, which can either buffer or exacerbate minority stress (Wallace et al., 2024). Future research using more nuanced, multi-level measures is needed to examine how family, school and community contexts interact to shape suicidality among LGBTQ+ youth.

Our findings collectively point to the need for systemic, multi-level interventions to address suicide risk among LGBTQ+ youth in Peru. Service delivery models must integrate mental health and social supports, with special attention to identity-affirming care, age-specific programming and outreach to underserved regions. In resource-limited settings such as Peru, feasible models for scaling such services include task-sharing approaches that train and supervise non-specialist providers within primary care, as reflected in the World Health Organization's (WHO) Mental Health Gap Action Programme (mhGAP), which includes guidance for assessment and management of self-harm and suicide in non-specialist settings (World Health Organization, 2023). When adapted with explicit LGBTQ+-affirming training and safeguards for confidentiality and safety, scalable, low-intensity psychological interventions deliverable by trained lay or non-specialist helpers (e.g., WHO's PM+) may also help expand access to evidence-informed support where specialist capacity is limited (Mwangala et al., 2024). Policy action is also needed to embed LGBTQ+ inclusion in mental health frameworks, enforce anti-discrimination protections and invest in workforce development. Future research should incorporate longitudinal designs to clarify causal relations, examine protective factors such as family acceptance and community connectedness and evaluate the effectiveness of targeted interventions. Regional variations in stigma and service availability also warrant exploration to inform locally tailored strategies.

This study leverages a large, national sample of LGBTQ+ youth across Peru, but several limitations must be noted. The cross-sectional design limits causal inference, and all data were self-reported, which may introduce recall bias or social desirability bias. In addition, analyses of barriers to mental health care were descriptive in nature and focused on youth reporting unmet need; we did not examine variation by age, gender identity or other subgroups, which represents an important area for future research. The online sampling approach may underrepresent LGBTQ+ youth with limited internet access or lower digital literacy, potentially excluding some of the most marginalized subgroups, including those in rural areas or in highly stigmatizing environments; additionally, recruitment via social media advertisements may have favored participation by youth who are more active on these platforms or more comfortable engaging with LGBTQ+-related content, potentially shaping the composition of the sample. Relatedly, youth experiencing higher levels of distress or unmet needs may have been more likely to enter and complete the survey, potentially inflating estimates of mental health symptoms and suicidality. We were unable to compare the included participants to those who were excluded or dropped out, limiting inferences about representativeness. In addition, the survey was limited to youth aged 14–24 years, leaving important gaps in evidence for pre-teen

populations whose developmental, social and emotional contexts differ substantially. Future research should address these gaps by incorporating diverse recruitment strategies, examining attrition patterns and complementing large-scale surveys with qualitative and mixed-methods approaches to capture the nuanced experiences of subgroups that may be systematically missed.

## Conclusion

This study documents an urgent and disproportionate burden of suicidality among LGBTQ+ youth in Peru and identifies multiple psychosocial and structural factors that are associated with suicidal ideation and attempts. Positive screens for depressive and anxiety symptoms, unmet mental health needs, younger age, gender diversity and greater outness were each associated with higher odds of suicidality in this cross-sectional sample. Addressing these risks will require coordinated efforts across health, education and policy sectors to expand access to identity-affirming, affordable and developmentally appropriate care. These results contribute to the global evidence base and reinforce the importance of culturally and contextually grounded suicide prevention strategies for LGBTQ+ youth in resource-limited settings. By integrating structural reform with community-based supports and developmentally tailored interventions, Peru has the potential to reduce suicide risk and improve the well-being of its LGBTQ+ youth population. Future longitudinal research is essential to clarify causal pathways and to determine how these factors interact over time to shape suicide risk.

**Open peer review.** To view the open peer review materials for this article, please visit http://doi.org/10.1017/gmh.2026.10169.

**Data availability statement.** The data that support the findings of this study are available from The Trevor Project. Restrictions apply to the availability of these data, which were used under a data-sharing agreement and are not publicly available.

**Acknowledgements.** We are deeply grateful to the community advisory board of LGBTQ+ youth in Peru, whose input and guidance shaped survey design, recruitment materials and interpretation of findings. We also thank The Trevor Project's Research Team for their partnership. Finally, we extend our sincere thanks to all the youth who participated in this study for their time, trust and insights.

**Author contribution.** Juan C. Jauregui led the conceptualization, analysis and writing of the manuscript. Michael Reyez-Diaz contributed substantially to the development of the methods section and provided critical feedback on analyses. Co-authors contributed to the study design, interpretation of findings and manuscript revisions. All authors reviewed and approved the final manuscript.

**Financial support.** Juan C. Jauregui was supported by the Fogarty International Center of the National Institutes of Health under Award Number D43TW009343 and the University of California Global Health Institute.

**Competing interests.** The authors declare none.

**Ethics statement.** The study was approved by the Comité Institucional de Ética en Investigación (CIEI) de Investigaciones Médicas en Salud (INMENSA) (No. 0012–2022 CIEI) in Peru and Solutions Institutional Review Board in the United States (Protocol Number 285).

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
