## [Reviewer Report]

Reviewer feedback

This manuscript makes a timely and significant contribution to the mental health field, investigating suicide risk among LGBTQ+ young people in Peru, drawing upon The Trevor Project’s 2024 national survey. The authors provide a valuable contribution and perspective from low and middle-income country, that it often overlooked in research papers. The paper therefore addresses a significant gap in the literature. The paper is clearly written, methodologically rigorous, and reflects a nuanced application of minority stress theory within the local sociocultural context, which I appreciated as a reader. Its findings hold important implications for public health, policy development, and mental health practice. I would recommend this manuscript for publication and have provided some brief comments/suggestions below for your consideration.

Strengths

There are a number of strengths that are worth mentioning. First, the use of a large national sample enhances the generalisability of the results, and allows for the meaningful subgroup analyses by age, gender identity, and mental health access. The analytical approach, including logistic regression models which have been controlled for key variables is both appropriate and transparent. The discussion section situates the findings within both global and regional literature, linking individual-level mental health factors to the broader structural determinants such as stigma, service accessibility, and discrimination. The authors have discussed the ethical considerations with care, particularly in terms of diverse identities and confidentiality. I also particularly appreciated the inclusion and integration of the community advisory board which has strengthened the overall study and the paper.

Areas for improvement

There are some areas that could be improved to enhance the manuscripts rigor and policy relevance. First, the cross-sectional nature of the data is appropriately acknowledged, but the discussion section could be expanded to emphasise the potential for reverse causation, particularly regarding the relationship between service use and suicidality. Second, although the authors noted that perceived community acceptance was not significantly associated with suicidality, it may be useful to explore any potential measurement limitations or indeed, moderating factors such as family acceptance, or school climate. Third, the barriers to care section may benefit from more quantitative detail, for example these could be linked to regression analyses or stratified by age or gender if chosen. Lastly, while the policy recommendations are strong, a brief mention of feasible intervention models in resource limited settings would strengthen this section and make for an interesting brief discussion.

Overall, this is a compelling paper that addresses a significant gap in the research, and pending any minor corrections, I would strongly recommend publication.

---

## [Reviewer Report]

This manuscript greatly contributes to the current literature on mental health among high-risk populations such as LGBTQ+. Furthermore, data on variables associated with suicide among LGBTQ+ youth is limited, particularly in low-and middle-income countries in Latin America. This manuscript nicely outlines the current needs and potential areas for intervention to improve mental health care for a vulnerable population in Peru. In addition to my comments above, please kindly see below a few remarks that aim to strengthen the manuscript:

1) On page 7, line 38, it is stated that recruitment took place through targeted advertisements on Facebook and Instagram. Could you please expand on the criteria used to identify the audience that received these advertisements?

2) On Page 11, lines 15-22, the sample characteristics are described, and Figure 1 is referenced. While Figure 1 outlines the reasons for excluding certain participants, it is unclear why some participants were excluded based on demographic requirements, as all regions/departments of Peru were included in the study. Could you please include this information in the manuscript?

3) On Page 9, lines 13-24, while PHQ-2 and GAD-2 are widely used screening tools, I suggest including a reference for these statements.

4) On page 9 and in the results section, it is inferred that a positive PHQ-2 and GAD-2 translates into depressive and anxiety symptoms. While a positive PHQ-2 and GAD-2 warrant further assessment to rule out an underlying mood or anxiety disorder, this does not necessarily mean that the person has depressive and anxiety symptoms. As such, the inference that clinically significant depressive and anxiety symptoms were measured in the manuscript, including Table 2, is not accurate, as participants did not have a clinical assessment. I suggest rewording parts of the manuscript where these statements and conclusions associated with them were included.

5) Based on comment 1 above, please include how the targeted advertisement might have impacted the selection of the surveyed individuals. Please include this in the paragraph where limitations are outlined.

6) On page 18, lines 40-47, it is stated that this study identified psychosocial and structural predictors of suicidality. This statement can be misleading, as it implies a predictive relationship and the ability to forecast a future outcome, which is not possible to conclude with this study (cross-sectional). Also, it is stated that “depression, anxiety, unmet mental health needs, younger age, gender diversity, and greater outness all emerged as significant risk factors”, which can also be misleading as a conclusion of this cross-sectional study, as this implies that exposure precedes the outcome in time. While literature supports these (depression, anxiety, unmet mental health needs…) as risk factors for suicidality through longitudinal studies, we cannot infer this from this study. Suggest editing this section to be consistent with the nature of the study.

---

## [Editor Report]

Dear Dr. Jauregui, 

I have read your manuscript and taken into consideration the reviewer comments. We would like to offer you the opportunity to address the reviewer comments in a major revision of your manuscript. Thank you for your interest in publishing your work in this special issue of Global Mental Health. 

Sincerely, 

Kristin Kosyluk, Guest Editor

---

## [Reviewer Report]

The authors addressed the feedback and comments provided in the first revision. This manuscript is articulate and compelling, and it greatly contributes to the current literature on mental health among high-risk populations such as LGBTQ+.

---

## [Editor Report]

Dear Dr. Jauregui and colleagues, 

I have had a chance to review your revised manuscript and your response to reviewers' comments. You have done a nice job addressing the reviewer’s concerns, and a previous reviewer agrees that their concerns have been addressed. We are pleased to inform you that your manuscript is now ready for publication in the special issue. Thank you for your important work, which we believe makes a novel contribution and adds to our understanding of suicide risk among LGBTQ+ youth in the Peruvian context. 

Kristin Kosyluk

Guest Editor